# Improvement in Biocompatibility and Biointegration of Human Acellular Dermal Matrix through Vacuum Plasma Surface Treatment

**DOI:** 10.3390/bioengineering11040359

**Published:** 2024-04-08

**Authors:** Ho Jik Yang, Byungchul Lee, Chungmin Shin, Boram You, Han Seul Oh, Jeonghoon Lee, Jinsun Lee, Se Kwang Oh, Sang-Ha Oh

**Affiliations:** 1Department of Plastic and Reconstructive Surgery, Chungnam National University Sejong Hospital, Sejong 30099, Republic of Korea; drhjyang@gmail.com; 2Department of Plastic and Reconstructive Surgery, Chungnam National University Hospital, Daejeon 35015, Republic of Korea; ontologiaa97@gmail.com (B.L.); piglet7474@gmail.com (C.S.); bry1021@naver.com (B.Y.); ginto9410@naver.com (H.S.O.); 3Plasmapp Co., Ltd., Giheungdanji-ro 24 Beon-gil, Giheung-gu, Yongin-si 17086, Gyeonggi-do, Republic of Korea; jhlee@plasmapp.com; 4Department of General Surgery, College of Medicine, Chungnam National University, Daejeon 35015, Republic of Korea; jinsunlee@cnuh.co.kr; 5Department of Emergency, College of Medicine, Chungnam National University, Daejeon 35015, Republic of Korea; 6Department of Plastic and Reconstructive Surgery, College of Medicine, Chungnam National University, Daejeon 35015, Republic of Korea

**Keywords:** vacuum plasma treatment, human acellular dermal matrix, biocompatibility, biointegration, reconstructive surgery

## Abstract

Efforts are ongoing to enhance the functionality of human acellular dermal matrices (hADMs), which are extensively utilized in reconstructive surgeries. Among these efforts, plasma treatments, particularly vacuum plasma treatments, have recently emerged in the medical field. This study aims to investigate the efficacy of a vacuum plasma treatment in enhancing the biocompatibility and biointegration of hADMs. Utilizing a plasma activator (ACTILINK reborn, Plasmapp Co., Ltd., Daejeon, Republic of Korea), hADMs were treated and evaluated through in vitro and in vivo analyses. Hydrophilicity changes were gauged by the blood absorption times, while SEM imaging was used to analyze physical surface deformation. Protein adsorption was measured with fluorescently labeled bovine serum albumin and fibronectin. For the in vivo study, mice were implanted with plasma-treated and untreated hADMs, and the post-implantation effects were analyzed through histological and immunofluorescence microscopy. The plasma-treated hADMs demonstrated a significantly enhanced hydrophilicity compared to the untreated samples. SEM imaging confirmed the maintenance of the microroughness after the treatment. The treated hADMs showed a significant reduction in fibronectin adsorption, a critical factor for cellular adhesion. In vivo, the plasma-treated hADMs exhibited reduced capsule formation and enhanced fibroblast infiltration, indicating improved biocompatibility and integration. These findings highlight the potential of a plasma treatment to enhance the performance of hADMs in clinical settings, offering a promising avenue for improving reconstructive surgery outcomes.

## 1. Introduction

The ideal foreign body used in clinical applications should induce minimal tissue reactions and exhibit long-lasting effects. Acellular dermal matrices (ADMs), first utilized in the 1990s for treating burns, have evolved as valuable tools in tissue reconstruction. ADMs, which are devoid of antigenic components to mitigate immune rejection, demonstrate a superior biocompatibility compared to synthetic soft tissue grafts [1,2,3].

Primarily employed in implant-based breast reconstruction, ADMs play a crucial role in covering the inferior breast pole. This facilitates the precise placement of tissue expanders or implants, shaping the new breast. The use of ADMs aids in implant positioning in the inframammary fold, mimicking the natural ptotic characteristics of soft tissues in the lower pole of the neo-breast. This results in implants that are more resistant to extrusion, less visible, and less palpable. ADMs also minimize implant-induced capsular contracture and appear to reduce associated inflammatory responses [3,4,5].

Despite their expense, ADMs are now commonplace in most implant-based breast reconstructions. As interest in ADM applications grew, scholars have explored their use in various surgical subspecialties over the past two decades. Commercial ADMs, varying in tissue origins and processing extents, have been developed, with human cadaver (hADM), bovine, and porcine tissues employed in different clinical scenarios [6]. ADM characteristics differ, including the tissue type, additives (e.g., antibiotics or surfactants), and preparation methods [5]. The ongoing efforts to enhance ADM performance include the use of mechanically made micronized ADMs [7]. In this way, efforts to improve the performance of ADMs have continued.

In modern medicine, cold atmospheric pressure plasmas find applications in the sterilization of medical devices and implants [8]. Their use in the treatment of viable tissues has also been explored, which has become a focus of medical research. Beyond therapeutic uses, cold atmospheric pressure plasma can facilitate surface modifications and biological decontamination. While research on the plasma treatment of biological materials is limited, no studies have addressed the plasma treatment of widely used ADMs clinically.

In this study, we aimed to evaluate the biocompatibility and biointegration efficiency of hADMs after a plasma treatment using a vacuum plasma discharge process that does not require the supply of additional gas, unlike atmospheric pressure plasma.

## 2. Materials and Methods

### 2.1. Device and Setup

#### 2.1.1. Plasma Device

To treat the hADM surface, we utilized a plasma activator (ACTILINK reborn, Plasmapp Co., Ltd., Daejeon, Republic of Korea) in conjunction with a dedicated holder. The holder consisted of a bottom, where the hADM could be placed, and a lid. When the holder with an hADM was loaded on the plasma activator, the holder was electrically connected to the ground of the plasma activator. When the plasma process began, the tube coming down from the top of the device contacted the bottom silicone cover of the holder’s seating part to form a chamber, thereby blocking the inside of the tube from external air. There was a vacuum port connected to the vacuum pump at the holder’s seating part, and through this, a vacuum of less than 10 torr was formed inside the tube and holder. In this vacuum state, a high-voltage power was delivered to the holder by the high-voltage power electrode present at the top of the vacuum tube. Plasma discharge occurred within the holder, uniformly exposing the hADM surface to the plasma (Figure 1).

#### 2.1.2. ADM Treatment Process

A freeze-dried, square, sheet-type hADM (Megaderm, L&C Bio Co., Ltd., Seongnam, Republic of Korea) measuring 10 × 10 × 1 mm underwent gamma irradiation sterilization. The plasma treatment involved loading the hADM into a dedicated holder, mounted on the plasma activator, and treating the hADM surface for 30 s. To treat the hADM surface, we utilized a plasma activator (ACTILINK reborn, Plasmapp Co., Ltd., Daejeon, Republic of Korea) in conjunction with a dedicated holder. The holder consisted of a bottom, where the hADM could be placed, and a lid. 

### 2.2. In Vitro Study

#### 2.2.1. Hydrophilicity Test of ADM Surface

Changes in the hADM surface’s hydrophilicity due to the plasma treatment were assessed by dropping 5 μL of defibrinated sheep blood (MB-S1876, KisanBio Co., Ltd., Seoul, Republic of Korea) on the dermal side of the hADM surface, and this was compared with an untreated hADM. The time to achieve the complete absorption into the hADM determined the hydrophilicity changes, measured by visually confirming that there were no droplets remaining on the hADM surface. Three samples of untreated and plasma-treated hADMs were used. Additionally, one hADM was divided into four parts and blood was dropped once on each part. That is, the complete absorption time of blood into the hADM surface for each plasma treatment condition was measured a total of 12 times.

#### 2.2.2. Evaluating Characteristics of ADM Surface Using Scanning Electron Microscope

Scanning electron microscopy (SEM, Thermo Fisher Scientific, Phenom XL, Waltham, MA, USA) was employed to visualize the physical changes of the hADM surface. Through SEM, the image surface topography was analyzed. In particular, the SEM images were used to compare the topographical differences before and after the plasma treatment.

#### 2.2.3. Protein Adsorption Experiment

To determine the protein interactions with the plasma treatment, bovine serum albumin (BSA, FITC conjugate Invitrogen™A23015, Waltham, MA, USA) and fibronectin (FN, Alexa Fluor™ 488 Conjugate Invitrogen™ F13191, Waltham, MA, USA) were used. Three samples of untreated hADMs and plasma-treated hADMs were prepared. First, the untreated hADMs and plasma-treated hADMs were soaked and hydrated with phosphate-buffered saline at 37 °C for 1 h. Optical and confocal fluorescence microscopes were used to observe the samples after protein adsorption.

### 2.3. In Vivo Study

#### 2.3.1. Experimental Animals

Eight-week-old male C57BL/6 mice (Samtako, Inc., Osan, Republic of Korea) were used in all the animal experiments. All the protocols were approved by the Animal Care and Use Committee of the Chungnam National University Hospital (CHNU-2023-IA0096) and adhered to the ethical guidelines of the National Institutes of Health and the International Association for the Study of Pain. The animals were randomly divided into control (*n* = 5) and experimental (*n* = 5) groups. The control mice received untreated hADMs, while the experimental mice received plasma-treated hADMs. We created 1.0 cm long incisions through the skin and panniculus carnosus, 0.5 cm from the medial dorsal line on the left side of the proximal dorsal region, and then prepared sub-panniculus pockets of 1.5 cm in diameter. In the control group, untreated hADMs were inserted into the pockets after being hydrated for 1 h. In the experimental group, plasma-treated hADMs were inserted after hydration. The mice were sacrificed at 4 weeks, and the hADMs and surrounding tissues were collected.

#### 2.3.2. Histological Analysis

The tissue sample of each mouse was fixed using 10% neutral-buffered formalin for three days, dehydrated in a series of % alcohol solutions, and embedded in paraffin. Each sample was then cut into 5 μm thick sections. Serial sections were mounted onto silicone-coated slides. Hematoxylin and eosin (H&E) and Masson’s trichrome stain (25088; Polysciences, Inc., Warrington, PA, USA) were used, and the samples were visualized under a light microscope and Panoramic MIDI II (3DHISTECH Ltd., Budapest, Hungary). The thickness of each capsule was measured as previously described and the mean thickness was calculated [9,10]. We calculated the cell penetration efficiency as the ratio of the hADM area occupied by fibroblasts to the total area of the hADMs on all slides. The area data were analyzed using Viewer ver.2.3 software (3DHISTECH Ltd., Budapest, Hungary).

#### 2.3.3. Immunofluorescence Microscopy

Immunofluorescence microscopy (Leica, Wetzlar, Germany) was used to examine the fibroblast and myofibroblast levels, and neovascularization in the capsule around the hADMs. The slides with hADMs were placed in sodium citrate (for antigen retrieval), boiled, and cooled for 30 min. The sides were incubated with antibodies against alpha-smooth muscle actin (α-SMA, 1:400, A5228; Sigma-Aldrich, St. Louis, MO, USA) and Vimentin-Alexa Fluor^®^ 488 Conjugate (1:200, #9854; Cell Signaling Technology, Inc., Danvers, MA, USA) overnight at 4 °C, and then with a biotinylated anti-mouse IgG secondary antibody (1:400, BA-2000; Vector) and a Cy3-streptavidin secondary antibody (1:400, PA43001; GE Healthcare) for 2 h at room temperature. The nuclei were stained with Hoechst 33342 (62249, Thermo Scientific™, Waltham, MA, USA) and then the slides were examined under a Leica DM2500 microscope.

### 2.4. Statistical Analysis

The statistical analyses utilized the GraphPad Prism ver. 9 software (GraphPad Software Inc., San Diego, CA, USA) with Student’s *t*-test. A *p*-value < 0.05 indicated significance. The data are presented as the means ± standard errors of the means (SEMs).

## 3. Results

### 3.1. In Vitro Study

#### 3.1.1. Hydrophilicity of hADM Surface before and after Plasma Treatment

As already reported in many previous studies, plasma treatments are known to increase the hydrophilicity of a surface. To confirm whether the plasma treatment successfully altered the wettability of the hADM surface, the wetting characteristics of the plasma-treated hADMs were tested and compared to untreated hADMs. The blood absorption time of the plasma-treated hADMs was significantly shorter than that of the untreated hADMs (*p* < 0.0001) (Figure 2). Therefore, we were able to confirm that the plasma treatment transformed the hydrophobic feature of the hADM surface into a highly hydrophilic characteristic.

#### 3.1.2. Surface Characteristics of hADMs before and after Plasma Treatment

Maintaining the microroughness on implant surfaces is crucial for cell adhesion and growth [11]. Therefore, when an hADM surface is treated with plasma, it is important not to harm this microroughness. The surface topography was observed through an SEM imaging surface. We imaged the exact location of the hADM surface before and after the plasma treatment. The SEM imaging of the hADM surface revealed no changes in the surface topography following the plasma treatment, indicating the preservation of the microroughness without any physical damage (Figure 3).

#### 3.1.3. Inhibition of Protein Adsorption by Plasma Treatment

To discern the cellular adhesion variations between the untreated and plasma-treated hADMs, a protein adsorption test was conducted with FITC-BSA-adsorbed samples and FITC-FN-adsorbed samples. We observed a reduced BSA adsorption intensity in the plasma-treated hADMs compared to the untreated hADMs, but this was not statistically significant. A significantly reduced FN adsorption intensity was observed in the plasma-treated hADMs (*p* < 0.01) (Figure 4).

### 3.2. In Vivo Study

#### 3.2.1. Changes in the Thickness of Capsules around Implanted hADMs

Both hADMs were well-tolerated by the mice, exhibiting biocompatibility without signs of infection or tissue rejection. The capsular thickness significantly differed between the control (ADMs, untreated hADMs) and experimental (ACTs, plasma-treated hADMs) groups. The capsules around the plasma-treated hADMs were much thinner than those in the control group at the 4-week mark, indicating that the plasma treatment prevented capsular formation. Significant differences were observed between the structures of the control and experimental samples (*p* < 0.01) (Figure 5).

The vimentin and α-SMA expression in the hADMs was evaluated via immunofluorescence microscopy. Vimentin+ cells (fibroblasts) were evident in both hADM groups, while the α-SMA+ cells (myofibroblasts) were not observed in either group, indicating active fibroblasts. Both groups exhibited successful engraftment (Figure 6).

#### 3.2.2. Fibroblast Infiltration and Proliferation in hADMs

In both hADM groups, fibroblast infiltration and proliferation were observed within the hADMs. In particular, fibroblast infiltration and proliferation occurred deeper into the hADMs in the plasma-treated hADM group (Figure 7A). In addition, the cell penetration efficiency was observed to be better in the plasma-treated hADMs compared to the untreated hADMs (Figure 7B). This was also statistically significant, indicating that fibroblast infiltration and proliferation were more vigorous in the plasma-treated hADMs (*p* < 0.05) (Figure 7C).

## 4. Discussion

ADMs are biotechnological tissues prepared from either human or animal skin. They have been widely employed in various medical applications, including breast reconstructions utilizing implants [3,4], abdominal wall and facial repair [2,12], pelvic reconstruction [13], and head-and-neck reconstruction [9]. The manufacturing of acellular dermal matrices (ADMs) involves the removal of cellular components that could potentially induce rejection or inflammation. The resulting tissue matrix retains its structural integrity, serving as a biological scaffold conducive to tissue ingrowth, angiogenesis, and, ultimately, tissue regeneration [6,7]. After mastectomy procedures, ADMs find applications in creating tissue expander pockets (in two-stage reconstructions) or implant pockets (in direct-to-implant reconstructions) [3]. Various ADM products, such as AlloDerm, Strattice, DermaMatrix, FlexHD, Permacol, and CG CryoDerm, differ in their tissue sources, manufacturing methods, storage, surgical preparation, size, and cost. However, clinical and preclinical studies have shown no significant differences in the complications or durability among these products [5,9,14].

Plasmas are gases that are partially ionized, comprising electronically excited atoms, molecules, ions, and free radicals. These highly reactive particles have the ability to rapidly introduce diverse chemical functional groups onto substrate surfaces [15]. Various types of plasma devices have been developed, particularly for industrial applications [16]. In particular, cold atmospheric pressure plasma has a surface-cleaning function that can remove contaminants such as biological and chemical agents, and as mentioned earlier, it can affect the characteristics of the surface by imparting chemical functions to the material’s surface [17]. Due to these characteristics, cold atmospheric pressure plasma can be used in dental fields for applications such as teeth whitening [18] and the inactivation of bacteria [19]. However, in the case of atmospheric pressure plasma, the injection of additional gas such as argon or nitrogen is usually required for stable discharge. Therefore, it requires the continuous management of gas. To overcome this weakness, a vacuum plasma process can be used in which the plasma is discharged at a low-pressure level without requiring additional gas. In the vacuum plasma treatment of dental implants made of titanium, the surface impurities were reduced and the hydrophilicity was increased, thereby increasing the biological activity, including improved protein and cell adhesion performance and cell differentiation [11,20].

This study investigated the effects of a vacuum plasma activator (ACTILINK reborn) on hADMs. The hydrophilicity of the hADM surfaces was measured through the blood absorption time. The blood absorption time of plasma-treated hADMs was significantly shorter than that of untreated hADMs (Figure 2). This indicates that the plasma treatment converted the hydrophobic nature of the hADM surface into a highly hydrophilic characteristic. SEM imaging of the surface topography post-plasma treatment revealed no changes, and no physical damage was observed (Figure 3). The increase in the hydrophilicity and the lack of physical damage to the plasma-treated hADMs is considered evidence of an improved biocompatibility and biointegration potential of the implant.

Protein adsorption on biomaterial and medical implant surfaces is crucial for biological reactions at the interface with the body. ECM proteins such as FN and BSA adsorb to the implant surface upon insertion, providing a substrate for cell adhesion, migration, and proliferation. Enhanced protein adsorption can contribute to increased foreign body reactions and capsular formation. The protein adsorption efficiency is an important indicator that demonstrates the biocompatibility of the implant [21]. Our results show a reduced BSA and FN adsorption intensity for plasma-treated hADMs compared to untreated hADMs. In particular, the FN adsorption intensity was statistically significant (Figure 4). Therefore, the plasma treatment of the hADM surface reduced the foreign body reaction, that is, improved the biocompatibility of the hADMs.

Capsular formation is a normal part of wound healing. The capsule isolates the wound from foreign bodies. Capsular formation is the result of a foreign body reaction. Therefore, reduced capsular formation indicates fewer foreign body reactions and a superior biocompatibility [22]. Our results show that the peri-hADM capsules in the plasma-treated hADMs were much thinner than those in the untreated hADMs, revealing that the plasma treatment of hADMs prevented capsular formation (Figure 5). Therefore, the present results suggest that the plasma treatment of hADM surfaces improves their biocompatibility.

Wound healing involves processes such as inflammation, angiogenesis, ECM deposition, host tissue remodeling, and the tissue integration of the ADMs. Effective integration is closely linked to long-term durability and resistance to infections. When an ADM degrades prematurely, before collagen deposition and neovascularization, the resultant tissue lacks the necessary thickness and tensile strength [23]. To determine the efficiency of biointegration, the infiltration and proliferation of fibroblasts into hADMs were measured. As a measurement method, we calculated the cell penetration efficiency as the ratio of the hADM area occupied by fibroblasts to the total area of the hADM on a slide [7]. In this study, it was confirmed that the cell penetration efficiency was significantly increased in the plasma-treated group, ensuring long-term structural integrity and durability, which indicates an increased biointegration efficiency.

Given that this study has not yet progressed to the clinical trial stage, its immediate clinical applicability may be limited. Furthermore, the absence of comparative experiments with other plasma treatment methods impedes a comprehensive understanding of the efficacy of a vacuum plasma treatment compared to alternative approaches. Therefore, additional research is warranted to address these gaps and further validate the findings of this study. Future investigations could include clinical trials to assess the feasibility and effectiveness of plasma-treated hADMs in actual surgical settings, as well as comparative studies to evaluate the relative benefits of different plasma treatment techniques. By conducting further research, we can gain a more thorough understanding of the potential benefits and limitations of plasma treatments for enhancing the performance of hADMs in reconstructive surgery, thereby facilitating their translation into clinical practice.

## 5. Conclusions

The application of a vacuum plasma treatment to hADMs was demonstrated with the aim of significantly improving their biocompatibility and biointegration. This innovative enhancement preserved the structural integrity of the hADMs while promoting favorable biological interactions that are crucial for successful reconstructive surgeries. The promise of plasma-treated hADMs may lead to better clinical outcomes, signifying a considerable advancement in regenerative medicine and tissue engineering techniques.

## Figures and Tables

**Figure 1 bioengineering-11-00359-f001:**
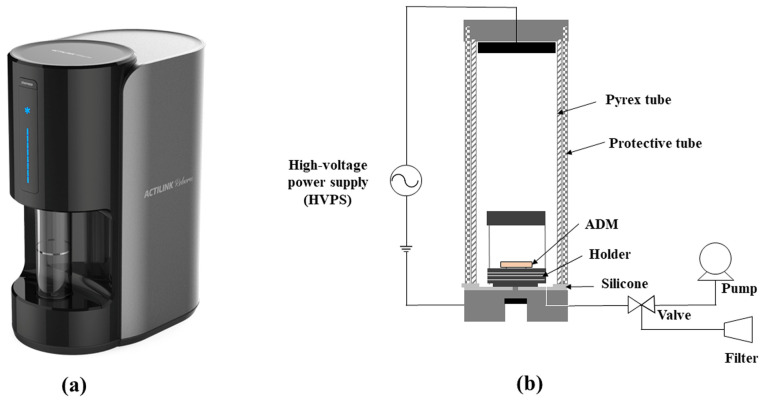
(**a**) Prototype view. (**b**) Schematic of the plasma activator (ACTILINK reborn).

**Figure 2 bioengineering-11-00359-f002:**
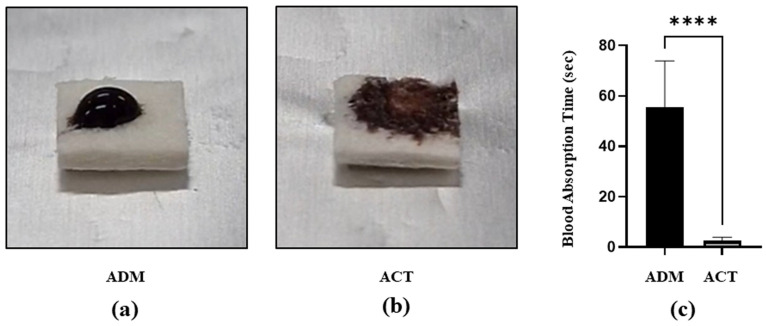
Hydrophilicity changes on hADM surface before and after plasma treatment. Representative images taken 2 s after dropping blood onto ADM. (**a**) Untreated (ADM) and (**b**) plasma-treated (ACT) hADMs. (**c**) Blood absorption time. Data are presented as means ± SEM (Student’s *t*-test; **** *p* < 0.0001; *n* = 12 per group).

**Figure 3 bioengineering-11-00359-f003:**
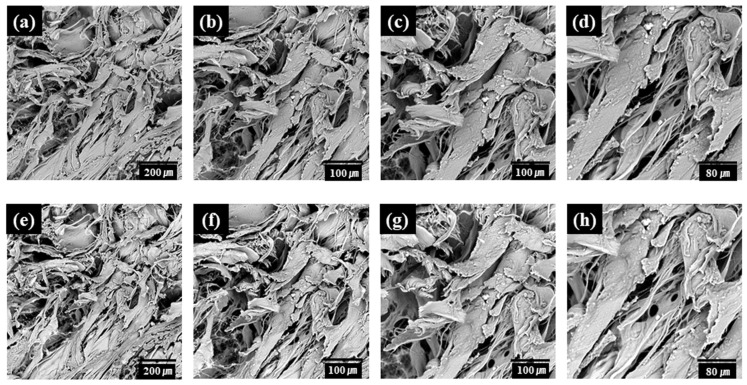
SEM images of hADMs before plasma treatment (ADMs) (**a**–**d**) and after plasma treatment (ACTs) (**e**–**h**).

**Figure 4 bioengineering-11-00359-f004:**
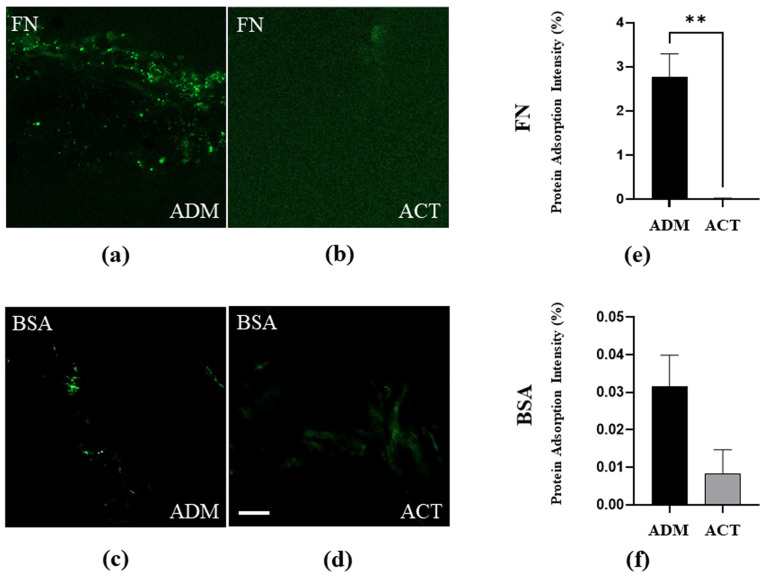
Fluorescence optical microscopy images of FITC-FN-adsorbed untreated (**a**) and plasma-treated (**b**) hADMs, and FITC-BSA-adsorbed untreated (**c**) and plasma-treated (**d**) hADM substrates. Scale bar = 20 μm. Calculated protein-binding area of FN (**e**) and BSA (**f**). Data are presented as means ± SEM (Student’s *t*-test; ** *p* < 0.01; *n* = 3 per group).

**Figure 5 bioengineering-11-00359-f005:**
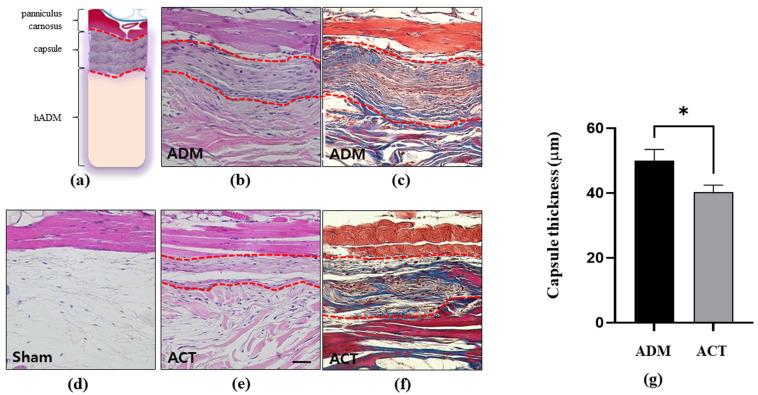
Thickness of peri-hADM capsule in illustration (**a**), sham (**d**), control (**b**,**c**), and experimental (**e**,**f**) groups. (A) Hematoxylin and eosin (H&E) and Masson’s trichrome (MT) staining of the plasma-treated, sub-panniculus hADMs (experimental group) and untreated hADMs (control group) at 4 weeks. Dot line is capsule. Scale bar = 50 μm. (**g**) Data are presented as means ± SEM (Student’s *t*-test; * *p* < 0.05; *n* = 5 per group).

**Figure 6 bioengineering-11-00359-f006:**
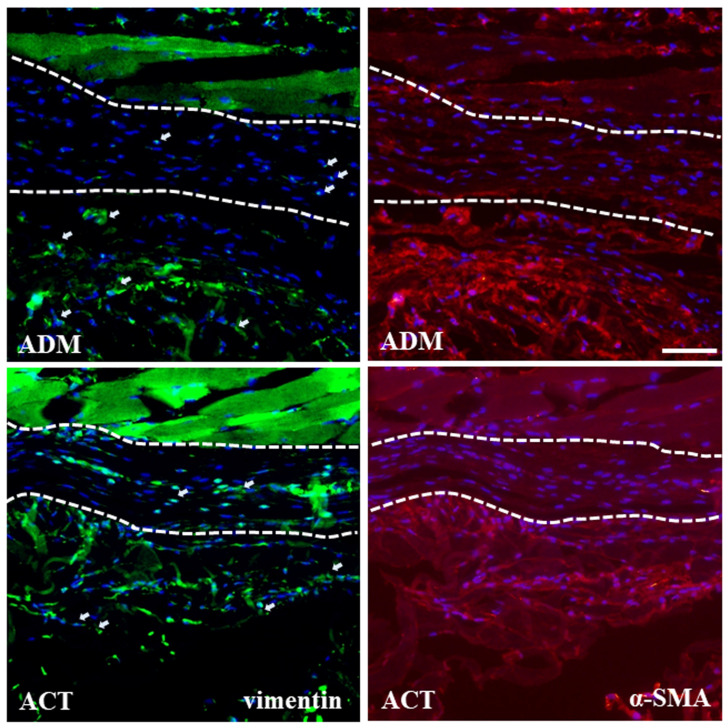
Immunohistochemical stainings of vimentin and α-SMA. Vimentin + cells (white arrow) were evident in the capsule and hADMs of both groups. α-SMA+ cells were not evident in the capsule or hADMs of either group. Between the dot lines is a capsule. Scale bars = 50 μm.

**Figure 7 bioengineering-11-00359-f007:**
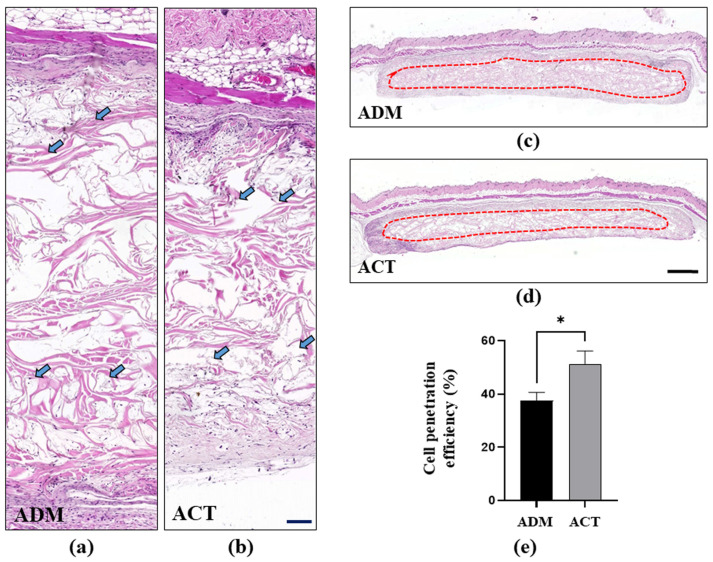
Fibroblast infiltration and proliferation inside hADMs in the experimental and control group. (**a**,**b**) Hematoxylin and eosin (H&E) staining of hADM sub-penniculus (scale bar = 100 μm). Fibroblast infiltration and proliferation into hADMs (arrow). (**c**,**d**) Lines with small dots show fibroblast locations (1000 μm). (**e**) Cell penetration efficiency as ratio of hADM area occupied by fibroblasts to total area of hADM (%). The data are presented as means ± SEM (Student’s *t*-test; * *p* < 0.01; *n* = 5 per group).

## Data Availability

All the data are presented in this manuscript.

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
