# Peer review of "Improvement in Biocompatibility and Biointegration of Human Acellular Dermal Matrix through Vacuum Plasma Surface Treatment"

_bioengineering, 2024, doi:10.3390/bioengineering11040359_

Round 1
Reviewer 1 Report
Comments and Suggestions for Authors
The authors present a very nice technological advancement paper titled Improvement of Biocompatibility and Biointegration of Human Acellular Dermal Matrix through Vacuum Plasma Surface Treatment, where they explore the effect of vacuum on plasma and its impact on the cells and tissues. The authors present that vacuum treating the plasma, it enhances its properties for multiple biological applications like cellular markers presentation and tissue structuring.
The authors presently nicely the changes in hydrophobicity of plasma with vacuum treatment and its effect on multiple biological parameters like proteins and cells. This is nice technological advancement. However the manuscript needs many modifications before it could be recommended for publishing.
1. Abstract - please modify by adding a line or two on what is the need for this process? what is the background.
2. How much is it technologically feasible to modify the existing equipment with vacuum and will it affect the cost of the device?
3. Are there any drawbacks like it could trigger some immune response in cells and blood?
4. The figures are not systematically arranged. Some of the figures could be merged to make them proper for publication.
5. How does this method compares with other similar technologies existing and what are the drawbacks of this method?
These above points should be clearly clarified.
Author Response
Dear Reviewer 1
RE: [Bioengineering] Manuscript ID: bioengineering-2948718 - Minor Revisions
Thank you for your email of May 25, 2024 informing us of the minor revisions of our manuscript. We have addressed the points raised by the referees as outline below. The referees’ comments are in italic followed by our responses. Additions and changes have been highlighted in yellow in the uploaded manuscript.
We appreciate your consideration regarding our manuscript and trust that it now meets the standards for publication.
With best regards,
Sang-Ha Oh
Comments to the Author
- Abstract - please modify by adding a line or two on what is the need for this process? what is the background.
Response: We appreciate the suggestion to strengthen our manuscript. In response to the reviewer’s feedback, we have added background information to the abstract.
Before.
Abstract: This study investigates the efficacy of vacuum plasma treatment in enhancing the biocompatibility and bio-integration of human acellular dermal matrices (hADMs) for reconstructive surgery applications.
After.
Abstract: Ongoing efforts to enhance the functionality of human acellular dermal matrices (hADMs) in reconstructive surgery have led to the exploration of novel techniques, including plasma treatment. This study aims to investigate the efficacy of vacuum plasma treatment in enhancing the biocompatibility and biointegration of hADM.
- How much is it technologically feasible to modify the existing equipment with vacuum and will it affect the cost of the device?
Response: Atmospheric pressure plasma devices can utilize plasma jets or specific chamber spaces, similar to our setup. Plasma jet systems are simpler but require additional accessories and maintenance items such as gas storage and connection pipes. In contrast, vacuum plasma systems, like ours, necessitate a vacuum pump, a valve, and a pressure sensor in addition to gas injection equipment. While vacuum plasma systems may incur slightly higher costs due to their complexity, they eliminate the need for additional gas-related accessories, potentially balancing out the overall expenses.
- Are there any drawbacks like it could trigger some immune response in cells and blood?
Response: Our study observed a significant reduction in capsule thickness in the plasma-treated group compared to the untreated group, suggesting a potential mitigation of immune reactions. This finding supports the notion that plasma treatment could reduce adverse immune responses to hADMs, thereby enhancing their biocompatibility and effectiveness in reconstructive surgery applications.
- The figures are not systematically arranged. Some of the figures could be merged to make them proper for publication.
Response: We have reorganized the placement of the figures as suggested.
- How does this method compares with other similar technologies existing and what are the drawbacks of this method?
Response: Previous research (Lee et al., 2022; Reference No. 20 in the manuscript) has demonstrated that vacuum plasma treatment generates higher energy plasma compared to atmospheric pressure methods, potentially leading to more efficient and rapid plasma effects. However, the increased complexity of vacuum plasma systems may result in slightly higher costs compared to simpler plasma jet systems.
The above content has been added to the manuscript.
Reviewer 2 Report
Comments and Suggestions for Authors
1- Figure 2 should be a,b and c because it contains 3 figures so a letter should be assigned to each figure.
2- Figure 3 contains 8 SEM images so they should take letter a-h. Resolution is very poor and scale bar is not obvious.
3- Figure 4 should be separated i.e. flouresence images alone should be 4 and take letters a-d then figures of protein binding of each substrate should be 5a and 5b.
4- BSA figure in figure 4 has no statistics.
5- Figures 5, 6, and 7 have the same problem of figure 4.
Comments on the Quality of English LanguageMinor revision is required
Author Response
Dear Reviewer 2
RE: [Bioengineering] Manuscript ID: bioengineering-2948718 - Minor Revisions
Thank you for your email of May 25, 2024 informing us of the minor revisions of our manuscript. We have addressed the points raised by the referees as outline below. The referees’ comments are in italic followed by our responses. Additions and changes have been highlighted in yellow in the uploaded manuscript.
We appreciate your consideration regarding our manuscript and trust that it now meets the standards for publication.
With best regards,
Sang-Ha Oh
Comments to the Author
1- Figure 2 should be a,b and c because it contains 3 figures so a letter should be assigned to each figure.
Response: Figure 2 has been appropriately labeled as suggested.
2- Figure 3 contains 8 SEM images so they should take letter a-h. Resolution is very poor and scale bar is not obvious.
Response: The SEM images in Figure 3 have been labeled a-h, and the scale bar has been adjusted for better visibility. High-resolution versions of Figure 3 have been provided as separate files.
3- Figure 4 should be separated i.e. flouresence images alone should be 4 and take letters a-d then figures of protein binding of each substrate should be 5a and 5b.
Response: Figure 4 has been divided into subfigures a-f as suggested.
4- BSA figure in figure 4 has no statistics.
Response: While a reduced BSA adsorption intensity was observed in plasma-treated hADM compared to untreated hADM, the difference was not statistically significant (p > 0.05). However, significantly reduced FN adsorption intensity was observed in plasma-treated hADM (p < 0.01).
5- Figures 5, 6, and 7 have the same problem of figure 4.
Response: Figures 5, 6, and 7 have been revised accordingly.
The above content has been added to the manuscript.
Reviewer 3 Report
Comments and Suggestions for Authors
The present Bioengineering-2948718 manuscript, by Ho Jik Yang et al, entitled "Improvement of Biocompatibility and Biointegration of Human Acellular Dermal Matrix through Vacuum Plasma Surface Treatment" refers to studies on the biocompatibility and bio-integration of human acellular dermal matrices (hADMs) for reconstructive surgery applications. The plasma-treated hADMs demonstrated significantly enhanced hydrophilicity compared to the untreated samples, whilst SEM imaging confirmed the maintenance of micro-roughness after treatment. With respect to the conducted in vivo studies, it was found that plasma-treated hADMs reduced capsule formation and enhanced fibroblast infiltration, indicating improved bio-compatibility and integration.
Both the in vitro and in vivo experiments were conducted by following the right methodology/protocols. The same holds true for the histological analysis, and the fibroblast, myofibroblast levels, and neovascularization in the capsule around hADM.
The article is concisely written, well documented and of interest to the cognizant reader. The findings presented herein suggest that plasma-treated hADM may lead to better clinical outcomes, signifying a considerable advancement in regenerative medicine and tissue engineering techniques.
Author Response
Dear Reviewer 3
RE: [Bioengineering] Manuscript ID: bioengineering-2948718 - Minor Revisions
Thank you for your email of May 25, 2024 informing us of the minor revisions of our manuscript. We have addressed the points raised by the referees as outline below. The referees’ comments are in italic followed by our responses. Additions and changes have been highlighted in yellow in the uploaded manuscript.
We appreciate your consideration regarding our manuscript and trust that it now meets the standards for publication.
With best regards,
Sang-Ha Oh
Comments to the Author
The article is concisely written, well documented and of interest to the cognizant reader. The findings presented herein suggest that plasma-treated hADM may lead to better clinical outcomes, signifying a considerable advancement in regenerative medicine and tissue engineering techniques.
Response: We appreciate the positive feedback and comments provided by the reviewer.
The above content has been added to the manuscript.
Reviewer 4 Report
Comments and Suggestions for Authors
The stud provides new portion of information on improvement of biological properties of human acellular dermal matrix through vacuum plasma surface treatment. The topic is sophsticated and the study design is proper.
Before further processing it will be nice to improved listed issues:
1.What was the age of the mice when they were sacrificed?
2.Tissue preparation for staining or microscopic imaging should be described in detail.
3.The same with Immunofluorescence and add information why these antigens were used.
4.Add study limitations.
Author Response
Dear Reviewer 4
RE: [Bioengineering] Manuscript ID: bioengineering-2948718 - Minor Revisions
Thank you for your email of May 25, 2024 informing us of the minor revisions of our manuscript. We have addressed the points raised by the referees as outline below. The referees’ comments are in italic followed by our responses. Additions and changes have been highlighted in yellow in the uploaded manuscript.
We appreciate your consideration regarding our manuscript and trust that it now meets the standards for publication.
With best regards,
Sang-Ha Oh
Comments to the Author
- What was the age of the mice when they were sacrificed?
Response: We utilized 8-week-old male C57BL/6 mice for all animal experiments and sacrificed them four weeks after concluding the experiment.
2.Tissue preparation for staining or microscopic imaging should be described in detail.
Response: We have expanded the description of tissue preparation methods, including surgical techniques and staining procedures.
Before:
Incisions were made, and sub-panniculus pockets prepared for hADM insertion.
After:
We created 1.0 cm long incisions through the skin and panniculus carnosus 0.5 cm from the medial dorsal line on the left side of the proximal dorsal region and then prepared sub-panniculus pockets 1.5 cm in diameter. In the control group, untreated hADM was inserted into the pockets after hydration for 1 hr. In the experimental group, plasma-treated hADM was inserted after hydration
Before:
Tissue samples underwent fixation, paraffin embedding, and sectioning for hematoxylin and eosin (H&E) and Masson’s trichrome staining.
After:
The tissue sample of each mouse was fixed using 10% neutral buffered formalin for three days and embedded in paraffin, dehydrated in series of % alcohol solutions and embedded in paraffin. The sample cut into 5um thick sections. Serial sections were mounted onto silicone-coated slides. Hematoxylin and eosin (H&E) and Masson’s trichrome staining (25088; polysciences, Inc. Warrington, USA) and visualized under a light microscope and Panoramic MIDI II (3DHISTECH Ltd, Budapest, Hungary).
3.The same with Immunofluorescence and add information why these antigens were used.
Response: Additional information on immunofluorescence procedures, including antigen selection rationale, has been included.
Before:
Antigen retrieval, blocking, and staining with alpha-smooth muscle actin (α-SMA) antibody were performed. Sections were examined under the Leica DM2500 microscope.
After:
The slides with hADMs were placed in sodium citrate (for antigen retrieval), boiled, and cooled for 30 min. Sides were incubated with antibody against alpha-smooth muscle actin (α-SMA, 1:400, A5228; Sigma-Aldrich, St. Louis, MO, USA) and Vimentin -Alexa Fluor® 488 Conjugate (1:200, #9854; Cell Signaling Technology, Inc. Danvers, MA, USA) overnight at 4°C and then with a biotinylated anti-mouse IgG secondary antibody (1:400, BA-2000; Vector) for 2 hr at room temperature and a Cy3-streptavidin secondary antibody (1:400, PA43001; GE Healthcare), for 2 hr at room temperature. The nuclei were stained with Hoechst 33342 (62249, Thermo Scientific™, Waltham, MA, USA) and then the slides were examined under the Leica DM2500 microscope.
4.Add study limitations.
Response: Given that this study has not progressed to a clinical trial stage, its immediate clinical applicability may be limited. Furthermore, the absence of comparative experiments with other plasma treatment methods impedes a comprehensive understanding of the efficacy of vacuum plasma treatment compared to alternative approaches. Therefore, additional research is warranted to address these gaps and further validate the findings of this study. Future investigations could include clinical trials to assess the feasibility and effectiveness of plasma-treated hADMs in actual surgical settings, as well as comparative studies to evaluate the relative benefits of different plasma treatment techniques. By conducting further research, we can gain a more thorough understanding of the potential benefits and limitations of plasma treatment for enhancing the performance of hADMs in reconstructive surgery, thereby facilitating their translation into clinical practice.
The above content has been added to the manuscript.